# Targeting Cancer Stem Cells to Overcome Chemoresistance

**DOI:** 10.3390/ijms19124036

**Published:** 2018-12-13

**Authors:** Toni Nunes, Diaddin Hamdan, Christophe Leboeuf, Morad El Bouchtaoui, Guillaume Gapihan, Thi Thuy Nguyen, Solveig Meles, Eurydice Angeli, Philippe Ratajczak, He Lu, Mélanie Di Benedetto, Guilhem Bousquet, Anne Janin

**Affiliations:** 1Institut National de la Santé et de la Recherche Médicale (INSERM), U1165, F-75010 Paris, France; toni.nunes@outlook.fr (T.N.); diaddin_h@hotmail.com (D.H.); christophe.leboeuf@univ-paris-diderot.fr (C.L.); morad.elbou@gmail.com (M.E.B.); guillaume.gapihan@gmail.com (G.G.); thuylinh.nong@gmail.com (T.T.N.); solveig.meles@edu.univ-paris13.fr (S.M.); eurydice.angeli@gmail.com (E.A.); philipperatajczak@gmail.com (P.R.); he.lu@inserm.fr (H.L.); melanie.dibenedetto@univ-paris13.fr (M.D.B.); 2Laboratoire de Pathologie, Université Paris Diderot, Sorbonne Paris Cité, UMR_S1165, F-75010 Paris, France; 3Hôpital de La Porte Verte, F-78004 Versailles, France; 4Université Paris 13, F-93430 Villetaneuse, France; 5Service d’Oncologie Médicale, AP-HP-Hôpital Avicenne, F-93008 Bobigny, France; 6Service de Pathologie, AP-HP-Hôpital Saint-Louis, F-75010 Paris, France

**Keywords:** cancer, chemoresistance, cancer stem cell, gold nanoparticles, functionalization, photo-thermal therapy

## Abstract

Cancers are heterogeneous at the cell level, and the mechanisms leading to cancer heterogeneity could be clonal evolution or cancer stem cells. Cancer stem cells are resistant to most anti-cancer treatments and could be preferential targets to reverse this resistance, either targeting stemness pathways or cancer stem cell surface markers. Gold nanoparticles have emerged as innovative tools, particularly for photo-thermal therapy since they can be excited by laser to induce hyperthermia. Gold nanoparticles can be functionalized with antibodies to specifically target cancer stem cells. Preclinical studies using photo-thermal therapy have demonstrated the feasibility of targeting chemo-resistant cancer cells to reverse clinical chemoresistance. Here, we review the data linking cancer stem cells and chemoresistance and discuss the way to target them to reverse resistance. We particularly focus on the use of functionalized gold nanoparticles in the treatment of chemo-resistant metastatic cancers.

## 1. Introduction

Cancer heterogeneity was demonstrated on clear-cell renal-cell carcinoma, using whole-genome analyses of multiple samples from the same primary tumor [1,2]. The mechanisms leading to cancer heterogeneity could be cancer stem cells or clonal evolution. According to the cancer stem cell theory, cancer stem cells, capable of self-renewal and differentiation, can generate mature differentiated cancer cells with genetic and epigenetic differences [3].

For 20 years, innovative medical treatments, including targeted therapies, anti-angiogenic drugs, and immunotherapies, have notably improved the prognosis of most metastatic cancers. However, secondary resistances almost constantly occur, and cancer stem cells are suggested as a potential source of this chemoresistance which has an increased risk of metastases and a lower survival rate [4,5,6,7,8].

In this narrative review, we analyze the recent literature on the role of cancer stem cells in chemoresistance to anti-cancer agents. We also provide a synthesis on how to target these cancer stem cells to reverse chemoresistance, for translational purposes.

## 2. Cancer Stem Cells and Chemoresistance

Two theoretical models are intended to explain the presence of cancer stem cells within a tumor. In the stochastic model, each cancer cell has the capability to dedifferentiate into a cancer stem cell, whereas in the hierarchical model cancer stem cells are the progenitors of differentiated tumor cells. According to this hierarchical model, cancer stem cells are able to self-renew and expand the cancer stem cell pool. They can also differentiate into heterogeneous cancer cell types to form the bulk of the tumor [9,10].

There is growing evidence that cancer stem cells are resistant to different types of stresses, including those generated by anti-cancer treatments [4,5,6,7,8,11], and, thus, could be associated with an increased metastatic risk and a lower survival rate [12].

Chemoresistance of cancer stem cells may be linked to: Their frequent quiescent state with a low proliferation rate, since most conventional cytotoxic agents target proliferating cells [7,13,14,15]. Cancer stem cells niches have been identified, where cancer stem cells may be quiescent and chemo-resistant. Depending on the cancer stem cell type, these niches may be hypoxic areas [6,7] or perivascular areas [16,17]. An innovative therapeutic perspective might be the use of hyperoxia to resensitize cancer stem-cells in the resistant metastases [18]. In chemoresistant glioblastoma cells, hyperoxia restores sensitivity to drugs [19]. In a murine model of breast cancer, hyperbaric oxygen treatment induced the mesenchymal-to-epithelial transition of cancer cells, restoring a more differentiated phenotype [20].The activation of drug-efflux mechanisms like ATP binding cassette (ABC) family transporters, especially ABCG2 [21] or the multidrug resistance P-glycoprotein (P-gp) namely ABCB1 (Table 1) [15,22,23]. Exposition to anti-cancer drugs, including taxanes, anthracyclines or antiangiogenic drugs, induces the expression of efflux pumps in cancer cells [24,25] and also in cancer stem cells leading to chemoresistance [26,27,28].The membrane expression of ABC transporters in cancer stem cells, especially ABCG2 and ABCB1, is currently used to identify them in the side population compartment [29,30]. A side population has been identified in many cancer types, based on their ability to efflux the lipophilic dye Hoechst 3342 [31,32,33,34]. To refine the selection of cancer stem cells within the side population, other stemness markers have been used. For example, in a preclinical model of ovarian cancer, side population and aldehyde dehydrogenase (ALDH)-expressing cancer stem cells have a greater tumorigenicity and are more resistant to cisplatin than the side population alone [35].The overexpression of DNA-repair mechanisms, including homologous recombination, non-homologous end joining [36,37], base-excision repair through increased poly (ADP-ribose) polymerase 1 (PARP1) activity [38], and decreased activity of programmed cell death [39,40,41]. These mechanisms are currently involved in resistance to anti-cancer drugs and radiation therapy. The concomitant inhibition of at least two DNA repair pathways is required to reverse chemo or radio-resistance. Typically, Breast Cancer *BRCA1* and *2* genes mutations lead to constitutive inactivation of homologous recombination. In metastatic ovarian and breast cancers with *BRCA* mutations, PARP inhibition with olaparib has been approved [42,43,44,45] and is associated with very high response rates when combined with cisplatin [46]. In a phase I study of radioresistant melanomas, concomitant inhibition of multiple DNA repair pathways restored sensitivity to radiotherapy [47]. To date, there are promising pre-clinical data on the benefit of specifically targeting DNA repair mechanisms in cancer stem cells [38,45,48,49,50].The acquisition of an epithelial-to-mesenchymal transition (EMT) phenotype. Cancer stem cells located at the invasive front of a tumor, contrary to quiescent cancer stem cells, have invasive and metastatic capabilities linked to an epithelial-to-mesenchymal transition phenotype [51]. In a large series of skin cancers, we have demonstrated that some cancer cells with an EMT phenotype also had stemness features and that they were preferentially distributed in the invasive front of the tumors [52]. In pre-clinical models, targeting epithelial-to-mesenchymal transition induces differentiation of cancer stem cells, reduces stemness and restores chemo and radiosensitivity [53,54,55,56,57].

Metastatic renal cancer samples offer the opportunity to study cancer heterogeneity and the role of cancer stem cells in resistance to treatments [1,2,6,58].

In pre-clinical studies, sunitinib, a leading anti-angiogenic drug, has been shown to mainly target neo-angiogenic micro-vessels, thus, inducing necrosis [6,59,60]. In clinical settings, there is also radiological evidence of necrosis induced by anti-angiogenic drugs among patients with metastatic renal cell carcinoma [61]. On cancer samples from patients with metastatic renal cell carcinoma, we showed that the numbers of cancer stem cells increased after treatment with sunitinib, but only in peri-necrotic hypoxic areas [6]. Using patient-derived xenografts from clear-cell renal cell carcinomas, we demonstrated that sunitinib was able to induce its own resistance by increasing the numbers of cancer stem cells in peri-necrotic hypoxic areas [6].

Our results are consistent with the clinical experience of cancer relapses after treatment with sunitinib [62], and with the identified two sub-types of renal cell carcinoma associated with resistance to sunitinib in patients. These sub-types are characterized by an activation of hypoxia pathways and a stem-cell signature [63]. So, sunitinib increases renal cancer stem cells numbers and contributes to its own resistance by its effects on endothelial tumor cells and the increase in cancer stem cells.

Regardless of tumor type, targeting tumor vessels could increase cancer stem cell numbers, because neo-angiogenesis is a mechanism common to all tumors [64].

We applied our experience on renal cancer stem cells to triple-negative breast cancers, a poor prognosis form of breast cancer in young women. On pre-treatment tumor biopsies of women with triple negative breast cancers, we have demonstrated that the numbers of breast cancer stem cells that were inversely correlated to response to chemotherapy were more numerous. We have also shown that these cancer stem cells were hypoxic, preferentially distributed in peri-necrotic areas, and in an autophagic quiescent state with autophagy features. Then, with our patient-derived xenograft models of triple-negative breast cancers, we demonstrated that drug resistance of autophagic cancer stem cells increased under hypoxic conditions, and we showed that inhibition of the autophagic pathway, and so cancer stem cells, was able to reverse the chemoresistance [7]. Our results present innovative therapeutic strategies to target cancer stem cells, and to overcome acquired resistance to anti-cancer drugs using multiple targets pathways simultaneously, namely autophagy and hypoxia.

Targeting cancer stem cells to reverse chemoresistance, thus, adds a new dimension to anti-cancer treatments, particularly for metastatic patients in resort situations.

## 3. Targeting Stemness Pathways to Overcome Chemoresistance

There are signaling pathways preferentially associated with cancer stem cells [65,66,67], including HEDGEHOG, NOTCH, STAT3, WNT/β-catenin, and NF-κB pathways that regulate stemness properties in many cancers (Table 2) [68].

The activation of the HEDGEHOG pathway is associated with cancer progression, acquisition of an EMT phenotype and cancer stem cell survival [69]. In tumor samples from patients with skin carcinoma, glioblastoma or colon cancers, HEDGEHOG is preferentially activated in cancer stem cells [69,70,71,72].

Janus Kinase/signal transducer and activator of transcription (JAK/STAT) are constitutively activated in tumor-initiating cells of patient-derived acute myeloid leukemia, and JAK inhibitors reduce their survival in vitro and their engraftment capability [73]. In patient-derived glioblastoma stem cells, STAT3 is strongly overexpressed and its inhibition decreases stemness properties and increases cell differentiation [74].

The NOTCH pathway is often implicated in tumorigenesis by promoting cell cycle progression, epithelial-to-mesenchymal transition, and apoptosis inhibition [75]. In patient-derived pancreatic xenografts, NOTCH genes are highly expressed in cancer stem cells. NOTCH inhibition decreases the numbers of pancreatic cancer stem cells in vitro and delays tumor engraftment and tumor growth [76]. In another model of the oesophageal cancer cell line and xenografts, NOTCH activity is correlated with stemness, tumor progression, and chemoresistance [77]. NOTCH inhibition restores radio-sensitivity of patient-derived glioma stem cells in vitro and impairs xenograft formation [78].

In addition to its role in maintaining stemness, WNT/β-catenin pathway is involved in EMT transition and metastatic process in many different cancers [76], including medulloblastoma, breast, gastric, and colorectal cancers [79]. In pre-clinical models of squamous cell carcinoma and breast cancer xenografts, blocking the WNT/β-catenin pathway decreases the numbers of cancers stem cells and enables treatment resistance to be overcome [80,81]. Cancer stem cell chemoresistance may be linked to the overexpression of ATP-binding cassette drug transporters, known as transcriptional targets of WNT/β-catenin [15,82].

NF-κB pathway constitutive activation is involved in many cancers and is also associated with aggressiveness features and cancer stem cell survival [88,89]. Pre-clinical studies using hematopoietic stem cells have demonstrated that loss of NF-κB activity impairs their stemness properties and their engraftment potential [90,91]. In breast cancer, cancer stem cells express higher NF-κB activity, and blocking NF-κB reduces stemness and metastatic potential in vitro and in vivo [92,93].

Targeting these stemness pathways in patients may thus be a way to reverse chemoresistance. It has provided some clinical benefit [94], but data are still lacking to make the link between response to an anti-cancer drug and the specific targeting of cancer stem cells [95].

Vismodegib is an inhibitor of the HEDGEHOG pathway, approved for the treatment of inoperable basal cell carcinoma of the skin [86,96]. In patients with metastatic gastric cancer, vismodegib did not lead to any clinical benefit except in a subgroup of patients with high levels of CD44 expressing cancer stem cells within the primary tumors [67,95]. In a phase II study of patients with progressive glioblastoma, vismodegib monotherapy was also disappointing, although it was able to decrease self-renewal capacities of glioblastoma-derived CD133-expressing stem cells [97].

Napabucasin is a STAT3 inhibitor with anti-tumor activity in vitro and in vivo in various cancers [98,99]. In association with paclitaxel, napabucasin did not lead to any benefit for the treatment of metastatic gastric cancer, probably due to the lack of patient selection. Subgroup analyses are ongoing to detect a benefit for patients with high levels of cancer cells expressing STAT3 [100,101]. In patients with advanced colorectal cancer, napabucasin led to a survival benefit, only in patients with tumors expressing phospho-STAT3 [102].

A limitation of targeting stemness pathways is that they are not specific of cancer stem cells and may particularly be activated in normal cells, leading to limiting toxic effects on normal tissues. Another limitation is the potential cross-talk between these pathways, an adaptive mechanism to maintain cancer stem cell survival. In a pre-clinical model of breast cancer, the inhibition of the PI3K pathway, another stemness pathway, increased the number of cancer stem cells through NOTCH activation [86,87]. Dual or multiple inhibitions of stemness pathways could be a way to overcome this limitation.

## 4. Targeting Cancer Stem Cell Surface Markers

Cancer stem cells can be identified by non-specific surface markers, usually dependent on the cancer type. Some markers have been proposed as preferential stemness markers (Table 3).

CD44, a hyaluronic acid receptor, is involved in numerous biological processes including cell adhesion, migration, drug resistance and apoptosis [150,151,152]. It has been identified as a stem cell marker in most cancer types, where it is associated with an invasive phenotype, metastatic potential [27,104,105,153,154,155,156], and chemoresistance [157]. In preclinical models of ovarian cancer, CD44-expressing cancer stem cells are more resistant to platinum salts and to paclitaxel than CD44-negative cells [27,106]. Knockdown of CD44 expression restores drug sensitivity to paclitaxel [158].

CD44 also has several isoforms of different functional significance. In pre-clinical models of colorectal cancer and in patient’s tumor samples, the variant 6 is highly expressed by cancer stem cells and interacts with WNT/β-catenin pathway leading to more aggressive tumors [159].

CD117 is the stem cell growth factor receptor, encoded by the c-KIT gene [114]. In preclinical studies, patient-derived CD117-expressing ovarian cancer stem cells have a high tumorigenic potential with features of chemoresistance to platinum salts and to paclitaxel. Using imatinib, a c-KIT/CD117 inhibitor, or anti-CD117 siRNA, enables the reversal of chemoresistance through Wnt/β-catenin pathway inhibition [27,82,115]. Unfortunately, in a phase II study of patients with metastatic recurrent and platinum-resistant ovarian cancer, imatinib alone did not lead to any benefit. This might be explained by the non-specificity of stem-cell markers within the same tumor [160].

CD133 is a common marker of cancer stem cells in patients, associated with a poor prognosis and resistance to conventional treatments [161,162,163]. In 131 patients with cancers of different types, a high level of CD133 mRNA expression in circulating mononuclear cells, including cancer stem cells, is associated with metastatic disease and worse survival [164]. In pre-clinical studies, patient-derived CD133-expressing ovarian cancer stem cells have increased engraftment capacities with chemoresistance to cisplatin [117,118]. In vivo, targeting CD133 efficiently inhibits the engraftment rate of various types of cancer stem cells [165,166,167]. In a pre-clinical model of glioblastoma, drug-conjugate bi-specific antibodies targeting CD133 increase drug delivery to glioblastoma stem cells with enhanced anti-tumor activity [168]. 

CD24 has also been largely studied as a cancer stem cell marker in many different types of cancers [104,111,126]. However, its role remains complex since both tumor-derived CD24-positive and CD24-negative cells may be chemoresistant with stemness properties, depending on the cancer type [126,169,170]. For example, in patients with ovarian cancer, CD24 expression is independently associated with an increased metastatic potential and a decreased survival [171]. By contrast, in patient-derived ovarian tumor samples, CD24-negative cells were more aggressive than CD24-positive cells, with stemness features and resistance to carboplatin and paclitaxel [172]. In hepatocellular carcinoma derived-xenografts, using a humanized monoclonal anti-CD24 antibody had anti-tumor effects, while there was no data on the eradication of the pool of cancer stem cells [173]. 

In patients, cancer stem cells usually have high levels of aldehyde dehydrogenase (ALDH) activity. This enzyme, by way of oxidizing aldehydes to carboxylic acids, may protect cancer stem cells and increase their chemoresistance by detoxification of anti-cancer drugs [174]. In a meta-analysis on 1258 patients with ovarian cancers, high ALDH expression in tumors is associated with decreased survival [129,175]. In preclinical models, ALDH-expressing ovarian cancer stem cells are chemo-resistant, and silencing ALDH gene expression enables chemoresistance to be reversed [176]. 

Epithelial cell adhesion molecule (EpCAM) is expressed in many types of cancer stem cells. EpCAM is involved in proliferation, migration, and invasion through WNT pathway activation [177]. In ovarian cancer xenografts, cancer cells co-expressing EpCAM and CD44 or CD24 had stemness properties with resistance to doxorubicin and cisplatin [137]. In patients with ovarian cancer, EpCAM expression correlates with aggressiveness and metastatic extent [178,179]. Using an anti-EpCAM therapeutic antibody showed promising results in xenografts models of colon and head and neck cancers [180]. 

Targeting cancer stem cells using monoclonal antibodies against surface markers still have limited clinical developments. 

Schlaak et al. reported the case of a patient with chemoresistant metastatic melanoma. He was successfully treated with an association of a cytotoxic agent and an anti-CD20 monoclonal humanized antibody to eradicate the CD20-expressing cancer stem cells in the melanoma metastases [181].

For renal cell carcinoma, several potential markers have been discussed, including CXCR4, CD105 or endoglin, and CD133 which correlate with poor prognosis [182,183]. Endoglin or CD105 is a membrane marker of tumor endothelial cells but has also been described as a potential cancer stem cell marker in renal cell carcinoma as well as other cancers [138,139,140]. Targeting CD105-expressing renal cancer stem cell inhibited tumor growth [141]. A phase I trial using an anti-CD105 antibody showed promising results, restoring sensitivity to anti-angiogenic agents in patients with metastatic renal cell carcinoma [184]. Various phase I trials have also been conducted in other cancer types with clinical benefits [185,186,187,188]. However, there is no published data regarding the effect of the anti-CD105 antibody on the pool of cancer stem cells. 

One limitation in targeting cancer stem cell markers is linked to the negative expression of some makers. Typically, breast cancer stem cells are CD44-positive and CD24-negative [112]. Anti-CD24 antibodies cannot, thus, be used as therapeutic agents. 

Cancer stem cells with different surface markers in a given tumor also contribute to tumor heterogeneity. We studied an exceptional clinical situation that of cancers in transplant patients. In a large series of skin cancers occurring in female patients with male kidney transplants, we studied donor-derived male cells using laser microdissection on tissue samples. An identical TP53 mutation in the tumor cells of a skin cancer and in tubular epithelial cells of the corresponding grafted kidney enabled us to demonstrate the participation of donor epithelial cells to the malignant epithelial proliferation in the recipient [189]. We also found that some epithelial cells, of male or female genotypes, had a cancer stem cell phenotype [52,190], thus, demonstrating the heterogeneity of cancer stem-cells within one and the same tumor.

To overcome this limitation, antibodies capable of dual or multiple inhibitions could be proposed.

## 5. Nanotechnologies to Overcome Chemoresistance

With cancer stem cells being more resistant to drugs than differentiated cancer cells, higher doses are required to kill them. Due to the need to limit toxicities on normal organs, this is not usually possible. Active drug delivery to cancer cells, including cancer stem cells, has thus been proposed using nanovectors.

Nanoparticles have been used for more than twenty years to deliver drugs to target cells, protecting them from degradation, with enhanced absorption, and improved distribution [191]. Targeting can be active, through recognition ligands, or passive as a result of enhanced permeability and retention in tumors, and the release of the drug in tumor cells by internalization [192]. 

A few nanovectors have been successfully developed for translational applications in cancer treatment, including liposomal doxorubicin for metastatic ovarian and breast cancers [193,194], and nanoparticle albumin-bound paclitaxel for metastatic breast and pancreatic cancers [195,196]. Liposomes were first described in the early seventies [197,198,199]. They are made of a copolymer of polylactic acid and polyglycolic acid [200,201], covered with polyethylene glycol (PEG) to decrease their captation by the reticuloendothelial system and, thus, increase their serum half-life [202]. In vitro, liposomal doxorubicin had a higher sensitivity than doxorubicin in chemoresistant CH LZ cells [203]. In patients with platinum-resistant ovarian cancer, liposomal doxorubicin had a low cardiac toxicity, enabling the use of very high doses, and, thus, delaying the occurrence of resistance to doxorubicin [204]. 

Active targeting of cancer cells can enhance drug delivery. This is done by functionalization of nanoparticles with specific antibodies [205,206,207,208,209,210]. One of the most explored targets is carcinoembryonic antigen (CEA) [210,211,212]. For active targeting of cancer stem cells and chemoresistance reversion, drug-loaded liposomes can be conjugated with a specific antibody, as successfully demonstrated in a pre-clinical model of glioblastoma [168].

Gold nanoparticles are promising nanovectors for translational purposes. Pure gold has two important properties. It is an inert bio-compatible chemical element, and so can be used for biomedical application; gold nanoparticles are non-toxic in pre-clinical models [213,214]. It has a unique optical and p lasmon surface resonance property, shifting the plasmonic resonance from 520 to 800–1200 nm wavelength, thus, converting light into heat [215]. Near infra-red radiation, with its peak absorbance wavelength in the 450 to 600 nm range, is transmitted through normal tissue with minimal absorption and can excite gold nanoparticles [216]. This produces focal hyperthermia [217].

Heating at supra-physiological temperatures from 40 to 47 °C leads to direct cell death in a time- and temperature-dependent manner [218]. This can be used to destroy cancer cells, and also cancer stem cells, thus, increasing the sensitivity of cancer to chemotherapy/radiotherapy-based treatments [11]. 

Hyperthermia, as an anticancer treatment, has already been discussed since the eighties [219]. Actually, it is used combined with intraperitoneal chemotherapy, at temperatures ranging from 40 to 45 °C, for the treatment of diffuse peritoneal carcinomatosis of ovarian or colon origin [220]. It is also used in radiofrequency ablation of metastases; its combination with chemotherapy, in colorectal liver metastases, results in a better overall control [221]. However, in these technologies, hyperthermia can be toxic due to heating of surrounding normal tissues. In photo-thermal therapy using plasmon surface resonance properties of gold nanoparticles, only irradiated cancer cells are heated [222], with no effect on surrounding normal tissues. In addition, the intense localized heat generated by irradiated carbon nanoparticles overcomes chemotherapy resistance in breast cancer cells [5].

Gold nanoparticles can be used in another promising application. They can be loaded with cytotoxic drugs for efficient delivery to cancer cells by passive distribution as malignant tumors have enhanced permeability and retention effect [223,224,225,226].

Active targeting can also enhance drug delivery. It was demonstrated by Pitsillides et al. that gold nanoparticles functionalized with an anti-CD8 can efficiently target T lymphocytes in vitro, and kill them after laser irradiation [222]. Functionalizing gold nanoparticles with cetuximab, an anti-EGFR therapeutic antibody, was able to enhance their internalization in cancer cells, while non-functionalized nanoparticles remained within the tumor stroma [208].

Another way to functionalize nanoparticles is with an anti-HER2 antibody; HER2 receptor is overexpressed in 15 to 20% of breast cancers [227] and usually maintained in resistance situations [228]; there are already three efficient anti-HER2 therapeutic antibodies trastuzumab, pertuzumab [229], and trastuzumab emtansine, a cytotoxic drug conjugated with trastuzumab to target resistant cancer cells [230]. Recently, it was shown, by Kubota et al., that gold nanoparticles conjugated with trastuzumab had a cytotoxic effect on HER2-overexpressing cancer cell lines, which was not the case for gold nanoparticles alone. They showed, in vivo, using HER2-overexpressing tumor xenografts, that tumor growth inhibition was linked to an autophagic state [231]. We have engineered a hybrid iron-gold nanoparticle conjugated with an anti-HER2 antibody for theranostic purposes. Active targeting of our hybrid gold nanoparticles on HER2-overexpressing breast cancer cells, in patient-derived xenografts of HER2-overexpressing breast cancer, increased gold delivery to cancer cells. Then the bulk of the cancer regressed, after pulsed-laser near-infrared irradiation, as a result of an anti-angiogenic effect alongside a direct cytotoxic effect on cancer cells [232].

## 6. Gold Nanoparticles Targeting Cancer Stem Cells to Reverse Chemoresistance

Functionalized nanoparticles with peptides or antibodies are currently being developed to actively target cancer stem cells, with translational relevance [233,234,235,236,237,238]. Gold nanoparticles coupled with a peptide recognizing CD133 have been used to target glioblastoma cancer stem cells, and as a contrast agent in a preclinical study [235]. Hyaluronic acid, a biocompatible linear polysaccharide with a high affinity to the CD44 receptor, is able to target cancer stem cells [233,239,240] since some CD44v isoforms are stemness markers [241]. PEGylated gold nanoparticles functionalized with an anti-CD44 antibody efficiently target breast or gastric cancer stem cells. Under electron microscopy, they are found in the cytoplasm of targeted cells within a few hours [234,236]. After intravenous injection of gold, nanostars conjugated with CD44v6 antibody in human gastric xenografts, near infra-red laser-irradiation significantly inhibits tumor growth [236]. 

The functionalization of gold nanoparticles with specific antibodies could enhance local drug delivery, and, combined with photothermal therapy, eradicate targeted chemo-resistant cancer stem cells (Figure 1).

A limitation of this approach could be the intrinsic resistance of cancer stem cells to hyperthermia, possibly related to the up-regulation of some heat-shock proteins. The fact that breast cancer stem cells are resistant to conventional hyperthermia in vitro, and to temperatures up to 47 °C [5], could explain the local recurrences of liver metastases following treatment by radiofrequency ablation [242]. According to mathematical modeling, the level of electromagnetic heat is limited at the interface between cancer and normal tissue [243]. To spare normal adjacent tissues, temperatures should not exceed 50 °C. Heating cancer stem-cells with carbon nanoparticle-mediated hyperthermia makes it possible to overcome resistance by generating intense sub-cellular localized heat [5], possibly above 50 °C [244]. Further experiments are required to check that targeting of normal stem cells within normal tissues adjacent to a tumor does not induce significant damages to these normal tissues.

Nanoparticles functionalized with molecules capable of dual or multiple inhibitions could be a way to overcome the problem of non-specificity of stem cell markers. For example, Zaimy et al. engineered complex nanoparticles to target hematopoietic stem cells in acute myeloid leukemia subtype 2. Their nanoparticles, loaded with five antisense oligonucleotides, are functionalized with an aptamer recognizing both CD33 and CD34 and are able to target AML-M2 cells in vitro [245].

In conclusion, the emergence of gold nanoparticle technology in the field of cancer biology will lead to significant theranostic progress with translational applications, particularly for metastatic patients in resort situation.

## Figures and Tables

**Figure 1 ijms-19-04036-f001:**
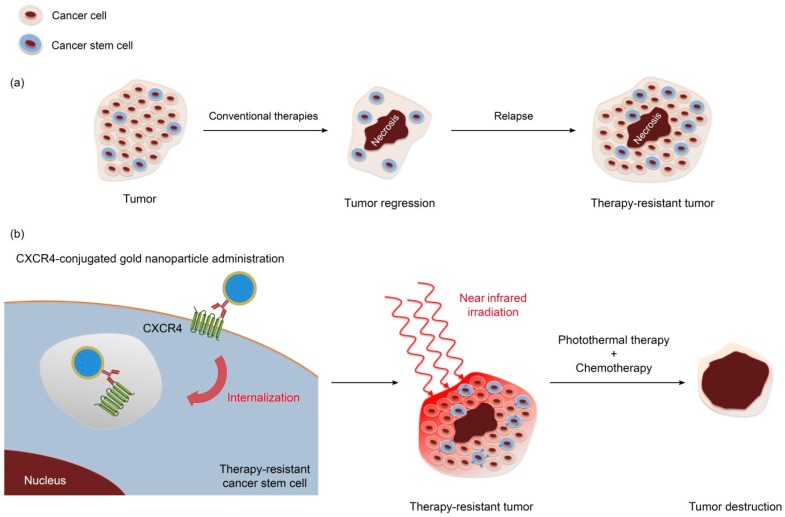
(**a**) Tumor response to conventional treatment is hampered by resistant cancer stem cells that can regenerate the tumor bulk. (**b**) Hybrid iron-gold nanoparticle conjugated with an anti-HER2 antibody and targeted against cancer stem cell marker CXCR4 for theranostic purposes; Photothermal irradiation destroys cancer stem cells and, thus, prevents tumor relapse following chemotherapy.

**Table 1 ijms-19-04036-t001:** ABC transporters involved in cancer drug resistance.

Gene	Chemotherapeutic Drugs Effluxed by Transporter	References
*ABCA1*	Cisplatin	[22]
*ABCA2*	Estramustine	[15]
*ABCB1*	Anthracyclines, actinomycin D, methotrexate, etoposide, mitomycin C, mitoxantrone, vincristine, vinblastine, taxanes, imatinib, nilotinib, EGFR TKI	[15,22,23]
*ABCB5*	Doxorubicin, 5-fluorouracil, camptothecin, mitoxantrone,	[22]
*ABCC1*	Anthracyclines, etoposide, camptothecins, methotrexate, mitoxantrone, vincristine, vinblastine, irinotecan, TKI as imatinib	[15,22,23]
*ABCC2*	Vinblastine, cisplatin, doxorubicin, methotrexate, paclitaxel	[15]
*ABCC3*	Cisplatin, doxorubicin Methotrexate, etoposide, vincristine	[15,22]
*ABCC4*	6-mercaptopurine, 6-thioguanine, methotrexate, topotecan	[15,22]
*ABCC5*	6-mercaptopurine, 6-thioguanine, and metabolites	[15]
*ABCC6*	Etoposide	[15]
*ABCC10*	Taxanes, vincristine, vinblastine, vinorelbine, cytarabine, gemcitabine	[23]
*ABCC11*	5-fluorouracil	[15]
*ABCG2*	Mitoxantrone, topotecan, anthracyclines, irinotecan, methotrexate, paclitaxel, TKI	[15,22,23]

ABC: ATP-binding cassette; EGFR: epidermal growth factor inhibitor; TKI: tyrosine kinase inhibitors.

**Table 2 ijms-19-04036-t002:** Cancer stem cells pathways.

Pathway	Functions	Cancers	References
HEDGEHOG	Regulates adult stem cells, tissue maintenance, and repair, EMT phenotype	Basal cell carcinoma, glioblastoma, medulloblastoma, rhabdomyosarcoma, colon cancer	[69,70,71,72]
JAK/STAT	Self-renewal properties in hematopoiesis and neurogenesis	Breast, glioblastoma, AML	[73,74]
NOTCH	Differentiation of stem cells and organ development	Breast, colon, pancreatic, prostate, skin cancers, CNS tumors	[75,76,77,78]
WNT/β-catenin	Self-renewal signal of stem cell and EMT phenotype	Melanoma, breast, gastric, colorectal, pancreatic, ovarian, skin cancers	[15,67,79,80,81,82]
PI3K/PTEN	Self-renewal and regulation of embryonic, hematopoietic, intestinal and neuronal stem cells, EMT phenotype	Glioblastoma, myeloproliferative disease, leukemia, breast cancer	[21,83,84,85,86,87]
NF-κB	Inflammatory and immune responses, proliferation, survival and differentiation, inhibit the activity of embryonic stem cell regulators SOX2 and NANOG	Hematologic, GI, Breast, GU, gynecologic, thoracic, head and neck cancers, fibrosarcoma, melanoma, squamous cell carcinoma	[88,89,90,91,92,93]

EMT: epithelial-to-mesenchymal transformation; AML: acute myeloid leukemia; CNS: central nervous system; GI: gastrointestinal; GU: genito-urinary.

**Table 3 ijms-19-04036-t003:** Markers preferentially used for the characterization of cancer stem cells.

Marker	Cancer	Phenotype	References
CD44	Ovary, stomach, breast, liver, head and neck, colon, prostate, pancreas	Tumorigenicity, spheroid formation, chemoresistance, hierarchical organization	[27,103,104,105,106,107,108,109,110,111,112]
CD117	GIST, ovary, skin, colon, blood, head and neck, sarcoma, germ cells tumors, prostate, lung, mesothelioma, breast, renal, CNS	Tumorigenicity, spheroid formation, self-renewal, chemoresistance, hierarchical organization, undifferentiated state	[27,111,113,114,115,116]
CD133	Blood, ovary, brain, pancreas, liver, skin, prostate, colon, lung, stomach, head and neck	Poorly differentiated gastric cancer, independent prognostic factor	[109,110,117,118,119,120,121,122,123,124,125]
CD24	Ovary, stomach, head and neck, pancreas	Tumorigenicity, self-renewal, hierarchical organization, chemoresistance	[104,110,111,123,126]
ALDH	Stomach, prostate, ovary, cervix	Tumorigenicity, phenotypical heterogeneity, chemoresistance	[35,123,127,128,129]
CD44/CD166/ALDH	Stomach, lung, colon, rectum	Tumorigenicity, chemoresistance, self-renewal	[110,123,128,130,131]
CXCR4	Stomach, blood, breast, ovary, melanoma, prostate, brain, lung, pancreas, colon, rectum, head and neck	Tumorigenicity, chemoresistance, angiogenesis, invasion	[132,133,134,135]
EpCAM	Stomach, ovary, pancreas	Tumorigenicity, phenotypical heterogeneity, self-renewal, metastasis, chemoresistance	[109,122,136,137]
CD105	Kidney, CNS	Proliferation, differentiation, migration, and angiogenesis, tumorigenicity	[138,139,140,141]
CD90	Stomach, kidney, CNS	Tumorigenicity, trastuzumab-reduced CD90-positive population	[120,123,138,139]
CD54	Liver, stomach, rectum	Metastases, tumorigenicity, spheroid formation, self-renewal	[122,142,143]
CD71-negative	Stomach	Quiescence, tumorigenicity, chemoresistance, tumor cell invasion	[144]
LGR5	Colon, liver, pancreas, stomach, brain, breast	Tumorigenicity, self-renewal, spheroid formation, self-renewal, invasion	[110,122,145,146]
Oct4	Stomach, head and neck, prostate, ovary, kidney, colon	Tumorigenicity, self-renewal, chemoresistance, hierarchical organization, invasion	[111,122,124,126,138,147,148]
Sox2	Stomach, head and neck, glioblastoma, kidney, brain, breast, pancreas	Well or moderately differentiated gastric cancer, tumorigenicity, self-renewal, chemoresistance, hierarchical organization	[35,111,122,125,138]
SP (efflux Vybrant^®^ DyeCycleTM Violet)	Ovary	Clonogenicity, asymmetric division and high tumorigenicity	[30]
SP (efflux Hoechst 33342)	Ovary	Chemoresistance, asymmetric division	[31]
SP (efflux Hoechst 33342)	Ovary	Chemoresistance	[149]
SP/ALDH^Br^	Ovary	Tumorigenicity, spheroid formation, pluripotency, chemoresistance	[35]

GIST: gastro-intestinal tumors; CNS: central nervous system; ALDH: aldehyde dehydrogenase; CSC: cancer stem cells; CXCR4: C-X-C chemokine receptor type 4; EpCAM: epithelial cell adhesion molecule; HCC: hepatocellular carcinoma; LGR5: Leucine-rich repeat-containing G-protein coupled receptor 5; Oct4: octamer-binding transcription factor 4; Sox2: sex determining region Y-box 2; SP: side population; Br: bright.

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
