# Peer review of "Targeting Cancer Stem Cells to Overcome Chemoresistance"

_ijms, 2018, doi:10.3390/ijms19124036_

Round 1

Reviewer 1 Report

In this manuscript entitled " Targeting cancer stem cells to overcome chemoresistance" by Tony nunes and colleagues, the authors have assessed the nanoparticle as key strategies to overcome chemoresistance by cancer stem cells. This is a well-designed manuscript and might have potential interest for the researchers.    

Author Response

We thank Reviewer 1 for his/her nice comment.

Reviewer 2 Report

Comment synopsis:  651 words

Manuscript ID ijms-402001, entitled “Targeting cancer stem cells to overcome chemoresistance” evaluated the application of gold nanoparticles for photo-thermal and Gold nanoparticles-conjugated antibodies for targeting cancer stem cells (CSCs), after reviewing the literature on CSCs as illustrated with 211 references and 3 tables plus one Figure.

The Manuscript appear to be systematic, impressive, of intriguing and interest.

Specific Comments and Suggestions for Authors (below) should be incorporated for its clarity, coherence, and logic flow.

Specific Comments, Questions, and Suggestions for Authors:

In Abstract: It’s too wordy. It should be rewritten to concise, straight to the point (fresh look).

Line 26: “Cancer stem cells can induce resistance to” –CSCs are resistant to treatment by nature, as pointed in Lines 48-50 with references #4-8 (That’s bad style of referencing – some details for each should be included. Lines 65-67 (native resistance) – as mentioned in Lines 68-82.

It’s confusing to back and forth, interchangeably, free of will to use these two terms “chemo-resistant cancer cells” and “chemo-resistant cancer stem cells” – Can you define and use them consistently?  For example, Line 24 (CSCs induced to resistance) and Line 34 (cancer stem cells and chemoresistance and discussed the way to target them to reverse resistance) – contradictory.

Abstract: “functionalized gold nanoparticles in the treatment” – can you elaborate more here as it’s your focus.

Lines 41-42: “Cancer heterogeneity was first demonstrated on clear-cell renal-cell carcinoma” – it’s not true as Cancer heterogeneity has been known for a long time.

Line 44: “mature cancer cells” “cancer stem cells” “cancer cells” – which is which? Can you define these terms and use them accordingly?

Lines 41-43 and Lines 55-56: So much repetitive in the sentence structure – paraphrase should be used.

Lines 59-64: “stochastic model” and “hierarchical model” – both are overlapped in the process. Some specifications should be added.

 Pages 2-3: Table 1 – references should be included to support.

 Lines 85-92: “membrane expression of ABC transporters in cancer stem cells, especially ABCG2” – that’s not CSC-specific, some clarification should be noted.

 Lines 102-103: “melanomas, concomitant inhibition of multiple DNA repair pathways restored sensitivity to radiotherapy” – How could that happen? You need to add the condition, such as “constitutive PARP activation” or “overexpression” – if not, side effects?

 Lines 121-122: “by increasing the numbers of cancer stem cells in peri-necrotic hypoxic areas [6].” – what’s the impact on normal tissues? Molecule biomarkers change (like Ref. #63?

 Lines 129-130: “Folkman J, Shing Y: Angiogenesis. J Biol Chem 1992, 267(16):10931-10934.” Did not specify CSCs in 1992. That’s wrong referencing.

 Lines 131-141: Under what comparison? What’s your normal control? How do you what’s the impact of targeting two pathways (normal vs. cancer)?

 Lines 44-179: “HEDGEHOG, NOTCH, PI3K, STAT3, WNT/β-catenin and NF-KB pathways”, also Table 2 – all of these are not CSC-specific. Targeting those can’t get you only CSC-specific effects. You mixed up CSC, cancer cells, with normal pathways.

 Table 3: These are not CSC biomarkers – you need to paraphrase to tie in with cancer conditions, e.g, mutations, overexpression, etc.

 Lines 306-310: “drug-loaded liposomes can be conjugated with a specific antibody” – Can you elaborate how you can target specific to glioblastoma?

 Lines 334-336: “gold nanoparticles functionalized with an anti-CD8 can efficiently target T lymphocytes in vitro, and kill them” – Why did you want to kill T cells?

 Lines 353-354 cannot cover references # [199-204].

 Lines 376-380: “According to mathematical modeling, the level of electromagnetic heat is limited at the interface between cancer and normal tissue [209]. To spare normal adjacent tissues, temperatures should not exceed 50°C. Heating cancer stem-cells with carbon nanoparticle-mediated hyperthermia makes it possible to overcome resistance by generating intense sub-cellular localized heat [5], possibly above 50°C [210].” Given the shared biomarkers, how could you differentiate the impact of CSCs from the normal cells? E.g., Figure 1b – CXCR4 is shared by normal stem cells and CSCs.

Author Response

Manuscript ID ijms-402001, entitled “Targeting cancer stem cells to overcome chemoresistance” evaluated the application of gold nanoparticles for photo-thermal and Gold nanoparticles-conjugated antibodies for targeting cancer stem cells (CSCs), after reviewing the literature on CSCs as illustrated with 211 references and 3 tables plus one Figure.

The Manuscript appear to be systematic, impressive, of intriguing and interest.

We thank Reviewer#2 for his/her nice comments.

Specific Comments and Suggestions for Authors (below) should be incorporated for its clarity, coherence, and logic flow.

Specific Comments, Questions, and Suggestions for Authors:

In Abstract: It’s too wordy. It should be rewritten to concise, straight to the point (fresh look).

To follow the advice by Reviewer#2, we have shortened the abstract which is now, page 1, lines 24 to 34: “Cancers are heterogeneous at the cell level, and the mechanisms leading to cancer heterogeneity could be clonal evolution or cancer stem cells. Cancer stem cells are resistant to most anti-cancer treatments and could be preferential targets to reverse this resistance, either targeting stemness pathways or cancer stem cell surface markers. Gold nanoparticles have emerged as innovative tools, particularly for photo-thermal therapy since they can be excited by laser to induce hyperthermia. Gold nanoparticles can be functionalized with antibodies to specifically target cancer stem cells. Preclinical studies using photo-thermal therapy have demonstrated the feasibility of targeting chemoresistant cancer cells to reverse clinical chemoresistance. Here, we reviewed the data linking cancer stem cells and chemoresistance, and discussed the way to target them to reverse resistance. We particularly focused on the use of functionalized gold nanoparticles in the treatment of chemo-resistant metastatic cancers.”

Line 26: “Cancer stem cells can induce resistance to” –CSCs are resistant to treatment by nature, as pointed in Lines 48-50 with references #4-8 (That’s bad style of referencing – some details for each should be included. Lines 65-67 (native resistance) – as mentioned in Lines 68-82.

In the introduction of the revised manuscript, we have now written line 25: “Cancer stem cells are resistant to most anti-cancer treatments”.

It’s confusing to back and forth, interchangeably, free of will to use these two terms “chemo-resistant cancer cells” and “chemo-resistant cancer stem cells” – Can you define and use them consistently?  For example, Line 24 (CSCs induced to resistance) and Line 34 (cancer stem cells and chemoresistance and discussed the way to target them to reverse resistance) – contradictory.

Cancer stem-cells but also cancer cells without any stemness features may be resistant to drugs. That is why we used these two different terms.

Lines 41-42: “Cancer heterogeneity was first demonstrated on clear-cell renal-cell carcinoma” – it’s not true as Cancer heterogeneity has been known for a long time.

Reveiwer#2 is right. We have now written in the revised manuscript, page 1, lines 39 and 40: “Cancer heterogeneity was demonstrated on clear-cell renal-cell carcinoma, using whole-genome analyses of multiple samples from the same primary tumor [References 1 and 2]”.

Line 44: “mature cancer cells” “cancer stem cells” “cancer cells” – which is which? Can you define these terms and use them accordingly?

In this sentence, we used the term “mature cancer cells” to talk about differentiated cancer cells generated by cancer stem cells.

To clarify this in the revised version of the manuscript, we have now written, page 1, line 43 “mature differentiated cancer cells”.

Lines 41-43 and Lines 55-56: So much repetitive in the sentence structure – paraphrase should be used.

We agree with reviewer#2 that the two first sentences of paragraph 2 were a repetition of the introduction. We have removed them in the revised version of the manuscript.

Lines 59-64: “stochastic model” and “hierarchical model” – both are overlapped in the process. Some specifications should be added.

The stochastic and the hierarchical models are clearly different, but we agree that if they co-exist, they necessary overlap. We had defined them as different models in the initial version of the manuscript, since we had written “In the stochastic model, each cancer cell has the capability to dedifferentiate into a cancer stem cell, whereas in the hierarchical model cancer stem cells are the progenitors of differentiated tumor cells.”

Pages 2-3: Table 1 – references should be included to support.

To follow the advice by Reviewer#2, we have now modified Table 1 and included references accordingly.

Lines 85-92: “membrane expression of ABC transporters in cancer stem cells, especially ABCG2” – that’s not CSC-specific, some clarification should be noted.

We do not understand this remark by Reviewer#2 since we did not write that membrane expression of ABC transporters is CSC-specific. Indeed, in the initial version of the manuscript, we had written, page 3: “The membrane expression of ABC transporters in cancer stem cells, especially ABCG2 and ABCB1, is currently used to identify them in the side population compartment”.

Lines 102-103: “melanomas, concomitant inhibition of multiple DNA repair pathways restored sensitivity to radiotherapy” – How could that happen? You need to add the condition, such as “constitutive PARP activation” or “overexpression” – if not, side effects?

This Phase I clinical study by Letourneau et al. is a very interesting study using an innovative therapeutic approach. This approach aims at inhibiting multiple DNA repair pathways using small DNA fragments of 32 base-pair mimicking DNA double-strand breaks. This molecule, named DT1, acts like bait for DNA-damage signaling enzymes by generating a ‘false’ DNA-damage signal. Then, there is no more possible recruitment of proteins involved in DNA-break repair at real damage sites.

The team of Marie Dutreix at Curie Institute has published robust pre-clinical data on the efficacy of these innovative molecules to overcome chemo and radioresistance in various types of cancers.

The Phase I study by Letourneau et al. is the first in-human application of these molecules, for the treatment of cutaneous metastases of radioresistant melanomas. In a total of 21 patients with 76 cutaneous metastases, objective response was observed in 60% of cases with 30% of complete responses, and no limiting toxicity. Indeed, DT1 molecules, while inhibiting several DAN-repair pathways concomitantly, seem not to have any toxic effect on normal cells.

Lines 121-122: “by increasing the numbers of cancer stem cells in peri-necrotic hypoxic areas [6].” – what’s the impact on normal tissues? Molecule biomarkers change (like Ref. #63?

Reviewer#2 raises an interesting question. To our knowledge, we do not know the tissue effect of sunitinib on normal tissues in humans. In patient-derived xenografts, using high doses of sunitinib, we observe diffuse endothelial toxicities with features of ischemic necrosis in kidney and bone marrow (data not shown). We do not know if, in normal tissue, these features are associated with the emergence of normal stem cells, like in cancer.

Lines 129-130: “Folkman J, Shing Y: Angiogenesis. J Biol Chem 1992, 267(16):10931-10934.” Did not specify CSCs in 1992. That’s wrong referencing.

Reviewe#2 is right, but we have cited this historical paper to tell that neoangiogenesis is a common feature to all cancers. To clarify the sentence, we have now added a coma in the revised manuscript, page 3, line 123 and 124: “Regardless of tumor type, targeting tumor vessels could increase cancer stem cell numbers, because neo-angiogenesis is a mechanism common to all tumors.”

Lines 131-141: Under what comparison? What’s your normal control? How do you what’s the impact of targeting two pathways (normal vs. cancer)?

We had conducted this study in our research unit [Reference 7 in the manuscript]: i) first, on pre-treatment tumor biopsies of women with triple negative breast cancers, demonstrating that the number of cancer stem-cells was inversely correlated to response to chemotherapy. We have also shown that these cancer stem cells were hypoxic, preferentially distributed in peri-necrotic areas, and quiescent with autophagy features; ii) second on patient-derived xenografts of metastatic triple negative breast cancers where we reproduced the results observed in patients. Using these pre-clinical models, we have also demonstrated that targeting autophagy pathway was associated to a decrease in numbers of cancer stem cells and to the reversion of chemoresistance to cytotoxic agents like cisplatin.

To answer to the questions by Reviewer#2, we have now reformulated this paragraph to make is clearer, page 4, lines 129 to 137: “We applied our experience on renal cancer stem cells to triple-negative breast cancers, a poor prognosis form of breast cancer in young women. On pre-treatment tumor biopsies of women with triple negative breast cancers, we have demonstrated that the numbers of breast cancer stem cells were inversely correlated to response to chemotherapy were more numerous. We have also shown that these cancer stem cells were hypoxic, preferentially distributed in peri-necrotic areas, and in a autophagic quiescent state with autophagy features. Then, with our patient-derived xenograft models of triple-negative breast cancers, we demonstrated that drug resistance of autophagic cancer stem cells increased under hypoxic conditions, and we showed that inhibition of the autophagic pathway, and so cancer stem cells, was able to reverse the chemoresistance [Reference 7].”

Lines 44-179: “HEDGEHOG, NOTCH, PI3K, STAT3, WNT/β-catenin and NF-KB pathways”, also Table 2 – all of these are not CSC-specific. Targeting those can’t get you only CSC-specific effects. You mixed up CSC, cancer cells, with normal pathways.

Reviewer#2 is right since none of these signaling pathways is specific to cancer stem-cells. They are also common signaling pathways in cancer cells without any stemness features, but also in some normal cells. However, they are preferentially activated and overexpressed in cancer stem-cells. That why we dedicated a paragraph on the possibility to specifically target these pathways to eradicate cancer stem cells and reverse chemoresistance.

However, due to their non-specificity, it is true that targeting these pathways may lead to toxic effects on normal tissues.

To follow the advice by Reviewer#2, we have added a sentence in the paragraph on targeting stemness pathways, page 5, lines 191 and 192: “A limitation of targeting stemness pathways is that they are not specific of cancer stem cells and may particularly be activated in normal cells, leading to limiting toxic effects on normal tissues.”

Table 3: These are not CSC biomarkers – you need to paraphrase to tie in with cancer conditions, e.g, mutations, overexpression, etc.

Reviewer#2 is true. Indeed, like for the pathways, the different surface markers that we are reporting here are not specific markers of cancer stem-cells since they may also be expressed by cancer cells without stemness features, but also normal cells.

To clarify this point, we have modified the title of Table 3, which is now in the revised version of the manuscript: “Markers preferentially used for the characterization of cancer stem cells”.

Lines 306-310: “drug-loaded liposomes can be conjugated with a specific antibody” – Can you elaborate how you can target specific to glioblastoma?

Kim et al. have demonstrated that dual-targeting drug-loaded immunoliposomes specifically deliver the drug to glioma stem cells by using angiopep-2 and anti-CD133 both conjugated to the immunoliposomes. Angiopep-2 is a peptide specifically recognizing a fragment of a lipoprotein receptor at the surface on endothelial cells of the blood-brain barrier. It serves as a Trojan horse, enabling the passage of the blood-brain barrier of these drug-loaded immunoliposomes conjugated with anti-CD133 antibody. Then anti-CD133 antibodies increase drug delivery to glioblastoma stem cells through an active targeting (Kim JS, J Control Release 2018).  

Lines 334-336: “gold nanoparticles functionalized with an anti-CD8 can efficiently target T lymphocytes in vitro, and kill them” – Why did you want to kill T cells?

In their study, Pitsillides et al. did the proof of concept that cells can be selectively targeted using gold nanoparticles conjugated with a specific antibody. Then, they have used the physical properties of gold to selectively heat targeted cells through photothermal therapy. They have used T cells as an example.

Lines 353-354 cannot cover references # [199-204].

In the manuscript we have submitted for review, we have combined these six references since they effectively refer to the same idea: “Functionalized nanoparticles with peptides or antibodies are currently being developed to actively target cancer stem cells, with translational relevance.” We have detailed some of the references in the rest of the paragraph.

Lines 376-380: “According to mathematical modeling, the level of electromagnetic heat is limited at the interface between cancer and normal tissue [209]. To spare normal adjacent tissues, temperatures should not exceed 50°C. Heating cancer stem-cells with carbon nanoparticle-mediated hyperthermia makes it possible to overcome resistance by generating intense sub-cellular localized heat [5], possibly above 50°C [210].” Given the shared biomarkers, how could you differentiate the impact of CSCs from the normal cells? E.g., Figure 1b – CXCR4 is shared by normal stem cells and CSCs.

This is a very interesting remark raised by Reviewer#2. We agree with Reviewer#2 that CXCR4 may be expressed by normal stem cells. For this reason, gold nanoparticles functionalized with an anti-CXCR4 antibody may target both CXCR4-expressing cancer stem-cells within a tumor but also normal stem-cells expressing CXCR4 in tissues adjacent to the tumor. Photothermal therapy of the tumor could theoretically induce damages to these normal tissues.

We do not believe this would be the case. Indeed, functionalized nanoparticles reach a tumor through two mechanisms: a passive mechanism called Enhanced Permeability and Retention (EPR) effect due to the abnormal neo-vascularization within a tumor, and an active targeting linked to the specific antibody used. The EPR mechanism by itself leads to a much higher delivery of drugs within a tumor compared to adjacent normal tissues.

However, this deserves to be confirmed. Typically, in our patient-derived renal cancer xenografts, this could be done by quantifying the number of normal and cancer stem cells expressing one marker of interest, here CXCR4. Then, after injection of functionalized gold nanoparticles and photothermal therapy of the tumor, we could assess the tissue and cell effects both in the tumor and in normal adjacent tissues, including necrosis and specific eradication of cells expressing CXCR4 in both compartments.

To follow the remark by Reviewer#2, we have now added a sentence at the end of the corresponding paragraph, page 12, lines 382 to 384: “Further experiments are required to check that targeting of normal stem cells within normal tissues adjacent to a tumor does not induce significant damages to these normal tissues.”

Reviewer 3 Report

In my opinion this is a clearly written review with just minor text mistakes that should be corrected before acceptance:

- throughout the paper "NF-KB" should be "NF-κB" ("k" latin character).

Author Response

We thank Reviewer 2 for his comment and we have now modified NF-κB throughout the manuscript and utilized the Latin character.